# An Overview of Aplysinopsins: Synthesis and Biological Activities

**DOI:** 10.3390/md21050268

**Published:** 2023-04-26

**Authors:** Eslam R. El-Sawy, Gilbert Kirsch

**Affiliations:** 1Chemistry of Natural Compounds Department, National Research Centre, Giza 12622, Egypt; 2Laboratoire Lorrain de Chimie Moléculaire (L.2.C.M.), Université de Lorraine, 57050 Metz, France

**Keywords:** aplysinopsin, sources, synthesis, analogs, bioactivity

## Abstract

Marine products are among the most promising sources of biologically active molecules. Aplysinopsins, tryptophan-derived marine natural products, were isolated from different natural marine sources including sponges, stony corals (hard corals) especially genus scleractinian, as well as sea anemone, in addition to one nudibranch. Aplysinopsins were reported to be isolated from different marine organisms related to various geographic areas such as Pacific, Indonesia, Caribbean, and Mediterranean regions. This review gives an up-to-date overview of marine alkaloid aplysinopsins: their various sources, their synthesis, and the fact that many aplysinopsin derivatives are biologically active compounds.

## 1. Introduction

Marine organisms produce thousands of organic compounds, many of which have an exceptionally high biological activity [1]. Aplysinopsins are tryptophan-derived marine natural products [2], whose name derives from the marine sponge *Thorecta aplysinopsis*. Aplysinopsin (**1**) was firstly isolated and chemically elucidated by Kazlauskas et al. [3]. Aplysinopsins have also been purified from other sponges, including *Verongia spengelii* [4], *Dercitus* sp. [5], *Hyrtios erecta* [6], *Smenospongia aurea* [7], *Thorectandra*, *Smenospongia,* and *Verongula rigida* [8,9]. Additionally, aplysinopsins were previously isolated from mollusk, *Phestilla melanobrachia* mollusk [10], and corals such as *Tubastrea coccinea* [10], *Tubastraea aurea* [11], *Dendrophyllia* sp. [12], *Tubastraea faulkneri* [13], and *Thorectandra* sp. [14].

Aplysinopsins possess an array of biological activities, such as, antimalarial [15,16], antimicrobial [16,17], monoamine oxidase (MAO) inhibitor [18], anti-depressant [19], and antiviral [20]. Recently, it has been shown that aplysinopsins act as a blood–brain barrier permeable scaffold for anti-cholinesterase and anti-BACE-1 activity (beta-site amyloid-precursor protein-cleaving enzyme 1) [21]. Additionally, aplysinopsin and its derivatives were found to possess significant anticancer activity against several cancer cell lines, including multidrug resistance (MDR) cell lines, leukemia, and breast, colon, and uterine cancer cell lines [22,23]. This review gives an up-to-date overview of marine alkaloid aplysinopsins: their origin, isolation sources, synthesis, analogs, and bioactivity.

## 2. Different Sources and Chemical Structures of Aplysinopsins

The chemical backbone of the natural aplysinopsins include a simple configuration of monomeric aplysinopsin-type structures and their brominated derivatives at the A ring, variation in the structure of the C ring, the presence and configuration of the C-8-C-1′ double bond, the oxidation state of the 2-aminoimidazoline fragment and *N*-alkylated at the B ring (Figure 1), in addition to the aplysinopsin dimers form.

Aplysinopsin, (*E*)-5-((1*H*-indol-3-yl)methylene)-2-imino-1,3-dimethylimidazo-li-din-4-one (**1**), was first isolated from the sponge genus *Thorecta* of the Australian Great Barrier Reef by Kazlauskas et al. [3]. Sequentially, aplysinopsin and its derivatives have been reported in many other marine organisms from various geographic areas (Table 1, Table 2, Table 3 and Table 4) [2].

## 3. Synthesis

The simple chemical structures of aplysinopsin motivated many researchers to prepare aplysinopsin-type structures in order to enrich various analogs and to study their activity besides the mode of action via their structure–activity relationships. Several synthetic approaches towards aplysinopsin-type structures have been reported.

The synthetic approaches toward the preparation of monomeric aplysinopsin-type structures involve the condensation reaction of the appropriate 3-formylindole **30** with a five-membered ring **31**, an imidazolidinone, by fusion [5,25] or boiling in acetic acid in the presence of sodium acetate [24] or boiling in piperidine [35] (Figure 1).

Molina et al. developed a simple and general entry to aplysinopsin alkaloids by tandem aza-Wittig/heterocumulene-mediated annelation [36]. Thus, ethyl 2-azido-3-(1-methylindol-3-yl)prop-2-enoate (**32**), available from 3-formyl-1-methyl indole (**31**) and ethyl azidoacetate, reacts with triphenylphosphine in dry dichloromethane to give the iminophosphorane (**33**). Compound (**33**), on the reaction with methyl isocyanate at room temperature, gave the carbodiimide (**34**). The reaction of (**34**) with ammonium acetate in acetonitrile afforded (**35**) in a 40% yield, while the reaction of (**34**) with methylamine in toluene at 45 °C afforded 2′-demethyl-3′-methylaplysinopsin (**36**) (Figure 2). On the other hand, the reaction of (**33**) with carbon dioxide in dry toluene at 90 °C afforded the isocyanate (**37**) in a 80% yield. The reaction of the isocyanate (**37**) with ammonium acetate in acetonitrile at room temperature afforded the urea derivative (**38**) (78%), which undergoes cyclization using acetic anhydride and provided the 3′-deimino-2′,4′-bis(demethyl)-3′-oxo aplysinopsin (**39**) in a 50% yield. Similarly, compound (**37**), by the reaction with methylamine and further cyclization by the action of acetic anhydride, led to the formation of (**41**) in a 50% overall yield (Figure 2).

Poisson et al. reported an alternative method to prepare aplysinopsin (**1**) (Figure 3) [37]. In detail, the reaction of dimethyl guanidine hydrochloride (**43**) with glyoxal (**42**) afforded the corresponding 2-(dimethylamino)-1,5-dihydro-4*H*-imidazol-4-one (**44**). The base catalyzed reaction of (**44**) with indole-3-carboxaldehyde (**30a**) yielded (*Z*)-5-((1*H*-indol-3-yl)methylene)-2-(dimethylamino)-1,5-dihydro-4*H*-imidazol-4-one (**45**). The methylation of (**45**) with methyl iodide, followed by basic hydrolysis using KOH (1N), provided aplysinopsin (**1**) (Figure 3) [37].

Skiredj et al. reported the synthesis of dictazole B (**29b**), an example of aplysinopsin dimers, as shown in Figure 4 [38]. After several trials and reaction optimization, Skiredj et al. reached the best route to obtain dictazole B via irradiation with artificial sunlight of the three monomers **1**, **10**, and **47**, in the presence of bismuth(III) triflate (Bi(OTf)_3_) (Figure 4).

Additionally, in 2014 and 2015, Skiredj et al. reported the total synthesis of the cycloaplysinopsin-type, tubastrindole B (**26**) (Figure 5), via a ring-expansion cascade of a dictazole-type precursor [39,40]. In detail, the cyclobutane (**48**) was produced by photochemical [2 + 2] homodimerization of aplysinopsin (**1**). Two minor compounds, **49** and **50**, have been isolated in addition to compound **48**. Compound (**48**) has been isolated by HPLC with a 16% yield to be involved in the next step. The ring-expansion of compound (**48**) in water with TFA under heating for 85 s and under microwave irradiation gave tubastrindole B (**26**) with a 40% yield (Figure 5).

In advanced research for the same research team, Skiredj and his group [41] have developed an unprecedented DNA-templated [2 + 2] photodimerization process for synthesizing spiro-fused cyclobutane-containing compounds including the natural heterodimer dictazole B (**29b**) (Figure 6). 

The method was developed based on the dimerization of (*E*)-aplysinopsin (**1**), which was previously shown to be unproductive in solution [39]. Upon this developed technique, exposure of 6-bromo-2′-*de*-*N*-methylaplysinopsin (**9**) and 6-bromoaplysinopsin (**10**), tryptophan-derived olefin, to light in the presence of salmon testes DNA (st-DNA) reproducibly afforded the corresponding homo-dimerized spiro-fused cyclobutane, Dictazole B (**29b**), with an excellent yield of 16% (Figure 6). Through this, compartmentalization properties offered by DNA could facilitate bypassing the inability of the monomers to self-organize in solution and ultimately promote the target cycloaddition.

## 4. Aplysinopsins Analogs

In 2009, various aplysinopsin analogs **51** and **52** were synthesized by the reaction of indole-3-carboxaldehyds **30**, namely, 5-bromo, 5-fluoro, and 6-brom-1*H*-indole-3-carboxaldehydes, with either creatinine or 2-imino-1,3-dimethyl-imidazolidin-4-one or 2-imino-1-methyl-3-ethylimidazolidin-4-one (Figure 7) [42]. 

The preparation of an *N*-benzyl-1*H*-indoles to be *N*-benzyl aplysinopsin analogs was of interest to a group of researchers as potential anticancer agents. In 2010, they reported a series of substituted (*Z*)-5-(*N*-benzylindol-3-ylmethylene)imidazolidine-2,4-dione analogs **54** using microwave irradiation and conventional heating methods (Figure 8) [23].

In 2011, they developed a series of substituted (*Z*)-5-((1-benzyl-1*H*-indol-3-yl)methylene)imidazolidin-2,4-diones **55** and (*Z*)-5-((1-benzyl-1*H*-indol-3-yl) methyl-ene)-2-iminothiazolidin-4-ones **56** utilizing microwave irradiation method (Figure 8) [22]. These analogs were evaluated for in vitro cytotoxicity against a panel of human tumor cell lines. Several of these analogs showed potent growth inhibition against melanoma UACC-257, OVCAR-8 ovarian, and MCF-7 breast cancer cells besides significant cytotoxicity. Therefore, these analogs could be regarded as useful lead compounds for further structural optimization as antitumor agents [22,23].

On the other hand, Cummings et al. reported low-molecular weight aplysinopsin analogs that served as a chemical scaffold for synthesizing compounds capable of differentiating between cloned human 5-HT_2C_ and 5-HT_2A_ receptor subtypes [43]. First, indole aldehydes **30** were reacted with the appropriate imidazolidinones to afford the corresponding aplysinopsin analogs **57** (Figure 9). Additionally, aplysinopsin analogs **58** were obtained via condensation of indole aldehydes **30** with thiohydantoin followed by methylation of intermediate thiones **58** to afford the corresponding thioethers **59**. Thioethers were allowed to react with methylamine to yield the corresponding *N*-methylamine aplysinopsin analogs **60** (Figure 9). The prepared analogs have been evaluated in the chick anxiety–depression model. In vitro receptor binding assays showed that analogs revealed high affinity with selectivity for 5-HT_2B_ over 5-HT_2A_ and 5-HT_2C_. Regarding the in vivo studies of aplysinopsins using different depression models, aplysinopsins were unable to reproduce the in vitro efficacy in an animal model. However, one compound, (*Z*)-2-amino-5-((5-bromo-1*H*-indol-3-yl) methylene)-1*H*-imidazol-4(5*H*)-one (**60a**), showed a modest antidepressant effect in the later stages of the evaluation period.

JakŠe et al. reported various aplysinopsin-thiohydantoin analogs via the reaction of ethyl 3-formyl-1*H*-indole-2-carboxylate (**61**) with the active methylene group of 2-thiohydantoin (**62**) in a mixture of acetic acid and sodium acetate (Figure 10) [44]. 

In 2016, Suzdalev and Babakova developed an efficient approach toward the synthesis of aplysinopsin analogs from 4-[(1*H*-indol-3-yl)-methylene]-1,3-oxazol-5(4*H*)-ones (**64**) [45]. The reaction of *Z*-isomers of oxazolones **64** with primary amines under reflux in dimethylformamide or ethylene glycol led to the formation of imidazolone derivatives **66** and **67** (Figure 11).

In 2021, Nuthakki et al. reported the insertion of sulphonamide moieties on the indole ring of aplysinopsin (**1**) and studied the effect of the obtained analogs on cholinesterases and beta-site amyloid-precursor protein-cleaving enzyme 1 (BACE-1) [21]. Aplysinopsin (**1**) inhibits electric eel acetylcholinesterase (AChE), equine serum butyrylcholinesterase (BChE), and human BACE-1 with IC_50_ values of 33.9, 30.3, and 33.7 μM, respectively, and it also showed excellent BBB permeability (8.92 × 10^−6^). The *N*-sulphonamide derivative **68b** displayed better cholinesterase inhibition and was found to effectively permeate the BBB (Pe > 5 × 10^−6^ cm/s). The sulphonamide derivatives were prepared by the reaction of aplysinopsin (**1**) with different substituted sulphonyl chlorides in the presence of 4-dimethyl-aminopyridine (DMAP) and *N*,*N*-diisopropylethylamine (DIPEA) (Figure 12) [21].

In 2021, Diederich and co-workers synthesized a series of aplysinopsin analogs EE (**71**) via the reaction of indole derivatives **69** with oxalyl chloride followed by the reaction with alkyl or arylamines (Figure 13) [35]. Through the in vitro and in vivo proliferation and viability screening of the newly synthesized aplysinopsin analogs on myelogenous leukemia cell lines and zebrafish toxicity tests, as well as the analysis of differential toxicity in noncancerous RPMI 1788 cells and PBMCs, they identified EE-84 (**71d**) as a promising novel drug candidate against chronic myeloid leukemia. Furthermore, they proved that EE-84 induced a senescent-like phenotype in K562 cells in line with its cytostatic effect. Finally, they demonstrated the synergistic cytotoxic effect of EE-84 with a BH3 mimetic, the Mcl-1 inhibitor A-1210477, against imatinib-sensitive and -resistant K562 cells, highlighting the inhibition of antiapoptotic BCL2 proteins as a promising novel senolytic approach against chronic myeloid leukemia [35].

Recently, El-Sawy et al. synthesized a new series of aplysinopsin analogs under the previous reaction conditions (Figure 14) and investigated their cytotoxic activity against prostate cancer [46]. Five analogs showed high antitumor activity via suppressing the expression of the anti-apoptotic gene Bcl2, simultaneously increasing the expression of the pro-apoptotic genes p53, Bax, and Caspase 3. The inhibition of BCL2 led to the activation of BAX, which in turn activated Caspase 3, leading to apoptosis. This dual mechanism of action via apoptosis and cell cycle arrest induction was responsible for the antitumor activity of the aplysinopsin analogs.

## 5. Biological Activity

Aplysinopsins possess an array of biological activities (Table 5).

## 6. Conclusions

Aplysinopsins are one of the most celebrated bicyclic arenas in the scientific literature, and they are tryptophan-derived marine natural products. Aplysinopsins represented the skeleton of indole and imidazolidin-4-ones cores that have been obtained from various sources of marine organisms. This survey touched upon the various sources of aplysinopsins, their synthesis, and the fact that many aplysinopsin derivatives are biologically active molecules. Aplysinopsins have been listed as anti-cancer, antimicrobial, inhibitors of monoamine oxidase (MAO), antidepressant, and serotonin receptors modulator. These facts open the field for scientific researchers to synthesize new aplysinopsin analogs to enrich their activity. An interesting observation is that the indole ring of aplysinopsin is considered an essential core of their structures. Therefore, future work should be focused on the optimization of the targeted imidazolidinone core.

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
