# Peer review of "An Overview of Aplysinopsins: Synthesis and Biological Activities"

_marinedrugs, 2023, doi:10.3390/md21050268_

Round 1

Reviewer 1 Report

Comments and Suggestions for Authors

Aplysinopsins constitute a family of marine natural products which offers structural diversity and variability to biological activity. The authors in this review give an idea of the importance of this family, describing the synthetic methods that have been implemented for the access to a large number of derivatives as well as their biological activity.

Some corrections related to the manuscript.

line 27. References 12, 13 and 14 are not related to the context.

Table 2. In the general structure what R1 stands for?

In the synthesis section all numbers of compounds should be in bold

Scheme 2 legend. In the manuscript you describe the reaction of 32 with Ph3P and in conditions (b) you wrote nBu3P. Which one is correct.

Also, conditions (d) erase carbodiimide.

conditions (d) and (g) are the same hence should be designated with the same letter. The same applies for conditions (h) and (j)

line 103 I think that dictazole B is number 29 and not 28. Same at Scheme 4.

From line 168 to the end of the manuscript. All compounds have wrong numbering. Please change accordingly.

Reference 18 Year is missing

Reference 47 Please add the necessary information.

Comments on the Quality of English Language

It is highly recommended for editing concerning English from a native speaker if possible.

Very few corrections.

line 95 catalysed

line 97 through basic hydrolysis

line 125 carboxaldehydes

line 132 of interest to a group of....

line 141 as useful lead compounds... (erase "a")

line 235-236 you could rephrase to: An interesting observation is that the indole.....

line 237 future work should be focused on the...

Author Response

Dear Professor

Editor, Marine drugs

Thank you for giving us the opportunity to resubmit our work (ID: marinedrugs-2356615) to published in your respectable journal.

My deep appreciation for the time and effort that you and the reviewers have dedicated to providing your valuable feedback on my manuscript. I have tracked the comments, and I responses all the comments during the manuscript.

Here is a point-by-point response to the reviewers’ comments.

Reviewer 1#

Line 27. References 12, 13 and 14 are not related to the context.

Response: The request has been checked and corrected

Table 2. In the general structure what R1 stands for?

Response: Thanks for pointing out our mistake. We checked and corrected by elimination the R1

In the synthesis section all numbers of compounds should be in bold

Response: Thanks, all compounds checked and corrected

Scheme 2 legend. In the manuscript you describe the reaction of 32 with Ph3P and in conditions (b) you wrote nBu3P. Which one is correct?. Also, conditions (d) erase carbodiimide conditions (d) and (g) are the same hence should be designated with the same letter. The same applies for conditions (h) and (j)

Response: Thanks for pointing out our mistake.

  • Condition (b) is Ph3P
  • For condition d): we deleted the (carbodiimide) as it refers to compound 34
  • For conditions d and g, they are not the same. The (d) run under heating for 45°C but (g) run under room temperature
  • For (h) and (j), we designated them with the same letter (h)
  • Finally we re-draw the scheme

Line 103 I think that dictazole B is number 29 and not 28. Same at Scheme 4.

Response: Thanks, the number checked and corrected.

From line 168 to the end of the manuscript. All compounds have wrong numbering. Please change accordingly.

Response: Thanks, the number checked and corrected.

Reference 18 Year is missing

Response: Thanks, checked and corrected

Reference 47 Please add the necessary information.

Response: Thanks, checked and corrected

Comments on the Quality of English Language

It is highly recommended for editing concerning English from a native speaker if possible.

Very few corrections.

line 95 catalyzed

line 97 through basic hydrolysis

line 125 carboxaldehydes  

line 132 of interest to a group of....

line 141 as useful lead compounds... (erase "a")

line 235-236 you could rephrase to: An interesting observation is that the indole.....

line 237 future work should be focused on the...

Response: Language corrections and polishing was performed

Regards

Reviewer 2 Report

Comments and Suggestions for Authors

The manuscript entitled “An overview of aplysinopsins: Synthesis and biological activities” gives an up-to-date overview of marine alkaloid aplysinopsins: various sources, synthesis, and biological activities. The topic seems to be interesting, while the manuscript seems to be incomplete. It should be revised carefully before considering to be published in Marine Drugs. The suggestions were listed below:

1.       Line 44, “Figure 1.” should be “Figure 1.”, and the author should name “Figure 1”.

2.       In table 4, “Tubastraea sp.” should be “Tubastraea sp.”

3.       Line 109, What is “hγ”? Is the author trying to say ""?

4.       Too many formatting errors, such as line 133, “(Z)-5-(N-benzylindol-3-ylmethylene)imidazolidine-2,4-dione”, while line 188: “N-sulphonamid”, line 192: “N,N-diisopropylethylamine”. The full review should remain uniform.

5.       Line 198, “in vitro and the in vivo”, while line 138 and line 158: “in vitro”.

6.       Line 187, “33.7 μM” should be “33.7 μM”, table 5, “1.7μM”, “3.5 μM”, and “0.33μM” should be “1.7 μM”, “3.5 μM”, and “0.33 μM”.

7.       Scheme 8, “2h”, “18 h” and “24 h” should be unified.

8.       There are a lot of errors in the format of the references, such as line 263, “J. Nat. Prod.”, line 306, “Journal of Natural Products” and line 310, “J Nat Prod”. The full review should remain uniform.

9.       Line 216 use “Bax” why line 217 use “BAX”?

10.   All compound numbers should be uniformly, the author could use “(numbers)”.

11.   Line 23, “et al”, while line 71, line 92, line 103, line 105, line 111, line 149 line 168, line 183 and line 212: “et al”.

12.   The review is incomplete, “DNA-Templated [2+2] Photocycloaddition: A Straightforward Entry into the Aplysinopsin Family of Natural Products“ reported the DNA-templated dimerization of Aplysinopsin family.

Comments on the Quality of English Language

The manuscript entitled “An overview of aplysinopsins: Synthesis and biological activities” gives an up-to-date overview of marine alkaloid aplysinopsins: various sources, synthesis, and biological activities. The topic seems to be interesting, while the manuscript seems to be incomplete. It should be revised carefully before considering to be published in Marine Drugs. The suggestions were listed below:

1.       Line 44, “Figure 1.” should be “Figure 1.”, and the author should name “Figure 1”.

2.       In table 4, “Tubastraea sp.” should be “Tubastraea sp.”

3.       Line 109, What is “hγ”? Is the author trying to say ""?

4.       Too many formatting errors, such as line 133, “(Z)-5-(N-benzylindol-3-ylmethylene)imidazolidine-2,4-dione”, while line 188: “N-sulphonamid”, line 192: “N,N-diisopropylethylamine”. The full review should remain uniform.

5.       Line 198, “in vitro and the in vivo”, while line 138 and line 158: “in vitro”.

6.       Line 187, “33.7 μM” should be “33.7 μM”, table 5, “1.7μM”, “3.5 μM”, and “0.33μM” should be “1.7 μM”, “3.5 μM”, and “0.33 μM”.

7.       Scheme 8, “2h”, “18 h” and “24 h” should be unified.

8.       There are a lot of errors in the format of the references, such as line 263, “J. Nat. Prod.”, line 306, “Journal of Natural Products” and line 310, “J Nat Prod”. The full review should remain uniform.

9.       Line 216 use “Bax” why line 217 use “BAX”?

10.   All compound numbers should be uniformly, the author could use “(numbers)”.

11.   Line 23, “et al”, while line 71, line 92, line 103, line 105, line 111, line 149 line 168, line 183 and line 212: “et al”.

12.   The review is incomplete, “DNA-Templated [2+2] Photocycloaddition: A Straightforward Entry into the Aplysinopsin Family of Natural Products“ reported the DNA-templated dimerization of Aplysinopsin family.

Author Response

Dear Professor

Editor, Marine drugs

Thank you for giving us the opportunity to resubmit our work (ID: marinedrugs-2356615) to published in your respectable journal.

My deep appreciation for the time and effort that you and the reviewers have dedicated to providing your valuable feedback on my manuscript. I have tracked the comments, and I responses all the comments during the manuscript.

Here is a point-by-point response to the reviewers’ comments.

Reviewer 2 #

  1. Line 44, “Figure 1.” should be “Figure 1.”, and the author should name “Figure 1”.

Response: Thanks for your suggestion. We checked and added title

  1. In table 4, “Tubastraea sp.” should be “Tubastraea

Response: Checked and corrected

  1. Line 109, What is “hγ”? Is the author trying to say ""?

Response: It means the reaction irradiation with artificial sunlight. Checked and corrected ().

  1. Too many formatting errors, such as line 133, “(Z)-5-(N-benzylindol-3-ylmethylene)imidazolidine-2,4-dione”, while line 188: “N-sulphonamid”, line 192: “N,N-diisopropylethylamine”. The full review should remain uniform.

Response:   All manuscript have been checked and corrected 

  1. Line 198, “in vitro and the in vivo”, while line 138 and line 158: “in vitro”.

Response:   All the manuscript have been checked and corrected 

  1. Line 187, “33.7 μM” should be “33.7 μM”, table 5, “1.7μM”, “3.5 μM”, and “0.33μM” should be “1.7 μM”, “3.5 μM”, and “0.33 μM”.

Response:   All abbreviation have been checked and corrected 

  1. Scheme 8, “2h”, “18 h” and “24 h” should be unified.

Response:   Checked and corrected

  1. There are a lot of errors in the format of the references, such as line 263, “ Nat. Prod.”, line 306, “Journal of Natural Products” and line 310, “J Nat Prod”. The full review should remain uniform.

Response:   checked and corrected

  1. Line 216 use “Bax” why line 217 use “BAX”?

Response:   the first one refer to gene, but the second refer to the protein

  1. All compound numbers should be uniformly, the author could use “(numbers)”.

Response:   checked and corrected

  1. Line 23, “et al”, while line 71, line 92, line 103, line 105, line 111, line 149 line 168, line 183 and line 212: “et al”.

Response:   checked and corrected

  1. The review is incomplete, “DNA-Templated [2+2] Photocycloaddition: A Straightforward Entry into the Aplysinopsin Family of Natural Products“ reported the DNA-templated dimerization of Aplysinopsin family.

Response Thanks for alerting us to what we missed during our search. The article added with reference 41.

Reviewer 3 Report

Comments and Suggestions for Authors

This review article describes the synthesis of aplysinopsins.  Generally, I do not find the content not very intellectually stimulating.  It is all focused on simple examples.  The authors need to add more structurally challenging examples to their manuscript.  

I found a few mistakes in manuscript:

a) In scheme 2, the structure of compound 34 is wrong.  For cvompound 37, the "N=C=O" group should be straight.

b) All the compound labels in the text should be in bold form.

c) In scheme 9, the amide bond of structure 83 looks odd.

I think this manuscript still need more work before it is publishable.

Comments on the Quality of English Language

The written English appears to be fine.

Author Response

Dear Professor

Editor, Marine drugs

Thank you for giving us the opportunity to resubmit our work (ID: marinedrugs-2356615) to published in your respectable journal.

My deep appreciation for the time and effort that you and the reviewers have dedicated to providing your valuable feedback on my manuscript. I have tracked the comments, and I responses all the comments during the manuscript.

Here is a point-by-point response to the reviewers’ comments.

Reviewer #3

  1. a) In scheme 2, the structure of compound 34 is wrong.  For compound 37, the "N=C=O" group should be straight.

Response: Thank you so much for pointing our mistake. Scheme 2 has been checked and corrected

  1. b) All the compound labels in the text should be in bold form.

Response: Thanks, the number checked and corrected.

  1. c) In scheme 9, the amide bond of structure 83 looks odd.

Response: Thanks, the structure of compound 83 checked and corrected.

I think this manuscript still need more work before it is publishable.

Response: Thanks, all manuscript has been checked carefully and improve

Round 2

Reviewer 3 Report

Comments and Suggestions for Authors

The authors have modified and improved the quality of this manuscript.  This work is now acceptable for publication.